# Spermatogonial Stem Cells in Fish: Characterization, Isolation, Enrichment, and Recent Advances of In Vitro Culture Systems

**DOI:** 10.3390/biom10040644

**Published:** 2020-04-22

**Authors:** Xuan Xie, Rafael Nóbrega, Martin Pšenička

**Affiliations:** 1Faculty of Fisheries and Protection of Waters, South Bohemian Research Center of Aquaculture and Biodiversity of Hydrocenoses, University of South Bohemia in Ceske Budejovice, Zátiší 728/II, 389 25 Vodňany, Czech Republic; psenicka@frov.jcu.cz; 2Reproductive and Molecular Biology Group, Department of Morphology, Institute of Biosciences, São Paulo State University, Botucatu, SP 18618-970, Brazil; rafael.nobrega@unesp.br

**Keywords:** spermatogonial stem cell (SSC), fish, spermatogenesis, florescence-activated cell sorting (FACS), magnetic-activated cell sorting (MACS), germ cell culture

## Abstract

Spermatogenesis is a continuous and dynamic developmental process, in which a single diploid spermatogonial stem cell (SSC) proliferates and differentiates to form a mature spermatozoon. Herein, we summarize the accumulated knowledge of SSCs and their distribution in the testes of teleosts. We also reviewed the primary endocrine and paracrine influence on spermatogonium self-renewal vs. differentiation in fish. To provide insight into techniques and research related to SSCs, we review available protocols and advances in enriching undifferentiated spermatogonia based on their unique physiochemical and biochemical properties, such as size, density, and differential expression of specific surface markers. We summarize in vitro germ cell culture conditions developed to maintain proliferation and survival of spermatogonia in selected fish species. In traditional culture systems, sera and feeder cells were considered to be essential for SSC self-renewal, in contrast to recently developed systems with well-defined media and growth factors to induce either SSC self-renewal or differentiation in long-term cultures. The establishment of a germ cell culture contributes to efficient SSC propagation in rare, endangered, or commercially cultured fish species for use in biotechnological manipulation, such as cryopreservation and transplantation. Finally, we discuss organ culture and three-dimensional models for in vitro investigation of fish spermatogenesis.

## 1. Overview of Fish Germ Cell Biology 

Spermatogenesis is a complex and orderly developmental process in which a single diploid spermatogonial stem cell (SSC) proliferates and differentiates to form mature spermatozoa. Spermatogenesis depends on the activity of the SSC, which can both self-renew to produce more stem cells or differentiate into daughter cells committed to spermatogenesis [1,2,3,4,5]. The proper balance between SSC self-renewal and differentiation is essential to assure the continuous homeostasis of spermatogenesis. The decision as to a SSC’s self-renewal or differentiation is mediated by cell–cell communication, and in vitro germ cell culture provides a novel platform with which to investigate the regulatory network that determines cell fate. Furthermore, germ cell culture can be combined with gene editing techniques such as clustered regularly interspaced shortpalindromic repeats (CRISPR)/CRISPR-associated(Cas) for germ-line transmission, cell transplantation, nuclear transfer, and in vitro spermatogenesis [6,7,8,9]. In the following section, we will review fish spermatogonial cell morphology, distribution, identification, and niche, and the endocrine and paracrine regulation of spermatogenesis. In the subsequent section, we will summarize the available protocols and advances in enriching undifferentiated spermatogonia according to their unique physiochemical and biochemical properties. Finally, we will review developments of traditional in vitro germ cell culture conditions to maintain proliferation and survival of spermatogonia in selected fish species, as well as organ culture and three-dimensional models for in vitro investigation of fish spermatogenesis. 

### 1.1. Spermatogenesis—an Overview 

Spermatogenesis is a continuous and dynamic developmental process which can be divided into three phases: the mitotic, or spermatogonial, phase with the generation of spermatogonia; the meiotic phase with primary and secondary spermatocytes; and the spermiogenic phase with the haploid spermatids emerging from meiosis and differentiating into motile, flagellated, haploid spermatozoa.

In the spermatogonial phase, the primary increase in germ cell numbers occurs during successive rounds of mitotic duplication of the spermatogonia. The number of spermatogonial generations, and hence the number of mitotic divisions before differentiation into spermatocytes, varies among, but not within, species. There can be as few as two (humans) and as many as 14 (guppy) generations, but, most commonly, five to eight generations are reported [10]. In this phase, in all invertebrates and vertebrates, at the end of mitosis, incomplete cytokinesis occurs, and the two newly-generated spermatogonia remain connected by a cytoplasmic bridge, instead of forming individual cells (Figure 1A,B). However, the cytoplasmic bridge is not present in the descendants of an SSC that enters a self-renewal pathway in which two single, isolated daughter cells are generated. Therefore, cytoplasmic bridges are considered a sign of SSC differentiation and are present during all subsequent germ cell divisions (Figure 1B). All differentiated descendants of an SSC form clones connected by cytoplasmic bridges through which their developmental steps are synchronized (Figure 1A). These bridges are cleaved when spermatogenesis is complete, and germ cells leave the germinal epithelium as spermatozoa [11].

During the meiotic phase, the spermatogonia differentiate into spermatocytes that go through two meiotic divisions characterized by reshuffling of the parental genetic material during the first division and the reduction to a haploid genome at the second division [10].

In the spermiogenic phase, the haploid spermatids emerging from meiosis differentiate into flagellated spermatozoa without further proliferation. The morphological changes in germ cells occurring during spermiogenesis involve reduction in cytoplasmic volume and organelles, maximum DNA condensation, and differentiation of the flagellum, and are similar among species. However, the final spermatozoon morphology can differ and sometimes provides taxonomic discrimination [10].

Following the general vertebrate scheme, the testes of fish are composed of the interstitial or intertubular compartment, and the germinal or tubular compartment, separated by a basement membrane [12]. The interstitial compartment contains the steroid-producing Leydig cells, blood/lymph vessels, macrophages, granulocytes, and connective tissue elements [13]. The peritubular myoid cells form a single layer of flattened cells surrounding the seminiferous tubules [14]. These cells are contractile and involved in the transport of spermatozoa and testicular fluid in the tubule [14]. The germinal compartment is composed of germ cells at various stages of development and their associated somatic Sertoli cells, which together form the germinal epithelium [12]. In the epithelium, germ cell survival and development depend on constant close contact with Sertoli cells [15]. Although many features are conserved in vertebrate spermatogenesis, the Sertoli/germ cell association differs among vertebrates. While anamniotes (fishes and amphibians) exhibit so-called cystic spermatogenesis, the amniotes (reptiles, birds, and mammals) present non-cystic spermatogenesis [12], in which the seminiferous epithelium can be divided into separate stages according to the cell associations observed in each tubular cross-section [16].

The germinal epithelium of amniote adult testes is composed of a fixed number of “immortal” Sertoli cells that support successive waves of spermatogenesis [15,16]. During these waves, a given Sertoli cell simultaneously supports several germ cell developmental stages (i.e., cells belonging to different germ cell clones). The Sertoli cell base may contact spermatogonia, with lateral segments contacting spermatocytes and early spermatids, and adluminal segments contacting late spermatids. In this type of spermatogenesis, Sertoli cells have been shown to be terminally differentiated in the testes, ceasing division during the early pre-pubertal phase, from approximately 10 and 20 days after birth in mice and rats, respectively [15,17]. However, recent studies have demonstrated that Sertoli cells from the transition region between the seminiferous tubules and the rete testis of adult testes remain undifferentiated for a longer period and are able to proliferate, although at a lower rate [17]. 

In anamniote vertebrates, the germinal epithelium is composed of spermatogenic cysts. The cyst as a morpho-functional unit is formed when a group of Sertoli cells envelope a single SSC [18]. As spermatogonia divide, the derived cells remain interconnected by cytoplasmic bridges [10,12,15]. Thus, the anaminiote Sertoli cell supports a single germ cell clone, while in amniote testes, depending on species, at least five germ cell clones at different stages of development are supported by a single Sertoli cell [10,12,15]. Sertoli cells from anaminiote testes are able to continuously proliferate, even after the onset of puberty [19]. 

The structural differences in the Sertoli/germ cell relationship of anamniotes and amniotes result in a less complex situation in fish than exists in mammals. Cystic spermatogenesis proceeds in a sequential manner, while, in non-cystic spermatogenesis, several processes occur simultaneously [15]. In vertebrates, the endocrine system has evolved as the master control system of spermatogenesis, and the somatic Sertoli, Leydig, and peritubular myoid cells became the primary targets for the major reproductive hormones, acting as paracrine relay stations for these signals in the testes [20,21].

### 1.2. Spermatogonial Stem Cell Niche 

Spermatogenesis relies on the activity of SSCs, which are capable of self-renewal to produce more stem cells or differentiation into daughter cells dedicated to spermatogenesis [1,2,3,4,5]. The balance between SSC self-renewal and differentiation is the basis of maintaining the homeostasis of spermatogenesis. If one process takes precedence over the other, testicular cancer (in case of self-renewal) or a depletion of spermatogenesis (in differentiation) is the outcome (Figure 2). 

The simultaneous process occurs in vertebrates showing continuous spermatogenesis, while, in seasonal breeding species, a switch from self-renewal to differentiation is observed as gonads begin to mature [10].

Spermatogonial stem cells are maintained in a specialized microenvironment in the testes known as the testicular niche (Figure 3) [2,3,4,22,23,24,25]. The niche provides growth factors and cell-to-cell interactions that regulate SSC activity in the testes: cell-cycle quiescence, maintenance of the undifferentiated state, proliferation via self-renewal, and apoptosis (Figure 3) [2,3,4]. 

The niche can be defined as a microenvironment that maintains the undifferentiated state of a stem cell by preventing its differentiation, and is usually composed of (1) supporting cells; (2) stem cells; and (3) the surrounding extracellular matrix (Figure 3) [26,27,28]. In mammals, SSCs lie on the basement membrane of the seminiferous epithelium and are in contact with Sertoli cells, which control the fate of the SSCs via physical and paracrine interactions [4,29,30,31]. In addition to Sertoli cells, peritubular myoid cells and Leydig cells may contribute soluble growth factors to the niche environment [32,33,34]. Mammalian SSCs are preferentially located in those areas of the seminiferous tubules near the interstitial tissue where Leydig cells and blood vessels reside (Figure 3) [33,35,36,37,38]. Recent studies have shown that type A undifferentiated spermatogonia are uniformly distributed on the basement membrane of seminiferous mouse epithelium [39]. It has been demonstrated that SSC self-renewal and proliferation are intensified in areas of high fibroblast growth factor (FGF), which corresponds to vasculature-proximal and interstitium-proximal regions, and SSCs must be exposed to a sufficiently high level of FGF in order to maintain the self-renewal state [40].

In anamniote species, SSCs and their niche remain poorly investigated. In fish and other anamniotes, SSCs are single cells, not lying directly on the basement membrane, but completely enclosed by Sertoli cell cytoplasmic extensions [10]. Similarly to mammals, fish have SSCs that are considered type A undifferentiated spermatogonia (A_und_) [1,10]. There is evidence for two sub-types of single, undifferentiated spermatogonia, type A_und*_ and type A_und_, in several fish species (Figure 4A–C), including ancient species, such as sterlet *Acipenser ruthenus* (Figure 4D–G) [10,22,41]. The A_und*_ spermatogonia exhibit a large nucleus with little heterochromatin; a high volume of the cytoplasm; a convoluted nuclear envelope; evident nucleoli; and particularly relevantly, darkly staining material near invaginations of the convoluted nuclear envelope [10,41]. These patches are known as “nuage,” material composed of ribonucleic acid (RNA) and RNA-processing proteins [10]. The A_und_ spermatogonia show a smooth nuclear envelope, darker chromatin with abundant patches of heterochromatin distributed in the nucleus, and less nuage [10,41]. 

These cells also exhibit differences in the cell cycle, as shown by bromodeoxyuridine 5-bromo-2’-deoxyuridine (BrdU), a marker of S-phase in pulse-chase experiments [22]. Nóbrega and collaborators [22], using BrdU pulse-chase, reported that BrdU was rapidly diluted in type A_und_ spermatogonia, while the percentage of BrdU-positive A_und*_ spermatogonia remained constant. The authors [22] suggested that type A_und_ constitute an “active” population with rapid proliferation and differentiation, as indicated by the more rapid loss of BrdU; A_und*_ are the “reserve” population with slow self-renewal proliferation, as indicated by relatively stable BrdU labeling. Two types of single A spermatogonia in humans display similar characteristics: a “pale” type acting as reserve, and a “dark,” active type [42,43]. In medaka *Oryzias latipes*, Nakamura and collaborators [44], using a BrdU pulse-chase experiment, also revealed distinct rapid and slow-dividing populations of oogonial stem cells. 

With respect to whether spermatogonia display a preferential distribution within the fish testes, studies of zebrafish [22] and *Astyanax altiparane* [45] have shown that both A_und_* and A_und_ spermatogonia are located near the interstitial compartment, in contact with androgen-producing Leydig cells and blood vessels (Figure 4A–C). These observations suggest that the androgens; growth factors; and vascular supplies of oxygen, nutrients, and hormones may play essential roles in SSC maintenance and self-renewal vs. differentiation in the fish testis niche [10,21]. 

Functional assays have been reported that consist of transplanting SSCs from a donor into a recipient testis in which endogenous spermatogenesis has been blocked [46]. Depending on self-renewal and differentiation capacities, transplanted spermatogonia are able to colonize the recipient testis and differentiate into functional gametes [46]. Studies have demonstrated donor-derived spermatogenesis following type A_und_ spermatogonia transplantation into recipient testes in several species of fish [22,41,47,48,49]. Type A undifferentiated spermatogonia have also been shown to exhibit sexual plasticity and the ability to de-differentiate and differentiate into oocytes when transplanted into zebrafish ovaries [22]. In addition, A_und_ spermatogonia transplanted into sexually undifferentiated larvae were able to differentiate into functional spermatozoa or eggs, depending on the genetic sex of the recipient [50,51]. Taken together, these findings provide evidence that SSCs represent a subset of type A_und_ spermatogonia.

Although available information on fish SSCs has expanded in recent decades, markers for SSCs have been identified in only a few species [52], presenting limitations to detecting and isolating SSCs. Similar to the situation in mammals, in the past decade, spermatogonium transplant was the only means of assessing the “stemness” of putative SSCs in fish [24,48,53,54,55]. 

There is evidence that promyelocytic leukemia zinc finger protein (PLZF), a transcription repressor essential for the maintenance of mammalian SSCs [56], can be a marker of SSCs in fish, as demonstrated in the rohu *Labeo rohita* [57], zebrafish [58,59], dogfish *Scyliorhinus canicula* [60], rainbow trout *Oncorhynchus mykiss* [61], and several species of catfish [9,62,63]. In the neotropical catfish *Rhamdia quelen*, in situ hybridization showed that *plzf* is strongly expressed in type A_und_ spermatogonia, but was also detected in type A_diff_ spermatogonia, although at less intensity [63]. 

Glial cell-derived neurotrophic factor (GDNF) is a Sertoli cell growth factor involved in mammalian SSC maintenance [64]. The GDNF-binding receptor GDNF family receptor alpha1 (GFRα1) attaches to the membrane of SSCs, and is considered a SSC marker in several mammalian species [64,65,66,67]. Recently, GFRα1 has been detected in type A_und_ of Nile tilapia *Oreochromis niloticus* [68], rainbow trout [69], dogfish [60], and common carp (Figure 5A). In rainbow trout, *gdnf* is expressed in germ cells from spermatogonia to spermatocytes but not in Sertoli cells, indicating that it is not secreted as an autocrine SSC niche factor in rainbow trout testes, unlike in mammals [69].

Transcription factors NANOG and Pou5f3 (POU family/Oct4) are expressed in SSCs of mammals, and in some fish, including medaka, zebrafish, Nile tilapia, and rainbow trout [61,68,70,71,72]. Both NANOG and Pou5f3 play important roles in the maintenance and self-renewal of undifferentiated and pluripotent cells [73,74]. In medaka gametes, *nanog* is expressed only in spermatogonia, being absent in somatic cells of ovary and testis [72]. In common carp, POU2 was detected in A_und_ spermatogonia, decreasing in expression as spermatogonia differentiated (Nobrega et al.—unpublished observations) (Figure 5B). 

Recently, an approach to establishing markers for SSC in fish has been developed [75]. The method consisted of generating monoclonal antibodies (mAb) to cell-surface molecules of rainbow trout type A_und_ spermatogonia by inoculating enriched live A_und_ spermatogonia into mice and screening with a combination of cell enzyme-linked immunosorbent assay, live-cell staining, and flow cytometry (FCM) [75]. Among the obtained antibodies, two (numbers 80 and 95) were capable of specifically labelling A_und_ spermatogonia of rainbow trout and zebrafish [75], while other antibodies (numbers 172 and 189) showed strong signals for type A spermatogonia and the oogonia of several species of salmonid [76]. By using these antibodies with fluorescence-activated cell sorting (FACS) [75] or magnetic-activated cell sorting (MACS) [77], it was possible to enrich A_und_ spermatogonia and increase transplant success rate in selected teleosts. This method presents the potential to identify molecular markers of SSCs in fish, and the potential to isolate and enrich SSCs for downstream studies, such as single-cell RNA-seq or in vitro experiments. Table 1 presents the major SSC markers currently reported in fish. 

### 1.3. Endocrine and Paracrine Regulation 

In vertebrates, pituitary gonadotropin follicle stimulating hormone (FSH) and luteinizing hormone (LH) control testicular development, and function by regulating the activity of local signaling systems involving sex steroids and growth factors [83,84], small RNAs [85], and epigenetic switches [86]. 

In rodents, FSH can modulate the production of Sertoli cell growth factors that are relevant for SSC self-renewal or differentiation [38]. Among the growth factors, the GDNF secreted by Sertoli cells plays an important role in SSC self-renewal [64], while activin A and bone morphogenetic protein 4 (BMP4), also produced by Sertoli cells, promote differentiation [87]. Growth factors produced by Leydig and peritubular myoid cells can also modulate SSC self-renewal or differentiation [32]. For example, colony-stimulating factor 1 secreted by Leydig and some peritubular myoid cells [32] stimulates SSC self-renewal. 

The gonadotropic hormones FSH and LH are important for testis development and spermatogenesis in fish [10]. Despite their similar roles in vertebrates, evolution has taken a different path for the gonadotropic hormones and their biological activities in teleost fish when compared to the other vertebrates. Leydig cells express not only the receptor for LH, typically seen in all vertebrates, but also the receptor for FSH [88,89,90,91]. Therefore, FSH can regulate Leydig cell functions, including stimulation of androgen [88,89,90] and production of growth factors acting on spermatogonial self-renewal and differentiation, such as Wnt5a [92] and insulin-like peptide 3 (Insl3) [93], respectively (Figure 6). Studies have reported elevated levels of circulating androgens and FSH coinciding with active spermatogonial proliferation in male Chinook salmon *Oncorhynchus tshawytscha*, and FSH stimulates spermatogonium proliferation in juvenile Japanese eel *Anguilla japonica* [88] as well as androgen production in several fish species [88,89,90,94]. Research in immature Japanese eels has shown FSH-induced spermatogenesis to be blocked by trilostane, a steroid hormone synthesis inhibitor, suggesting that FSH effects were mediated by androgens [88]. On the other hand, studies in zebrafish have demonstrated the impact of FSH on germ cell development independent of androgens [20]. This is supported by evidence that hundreds of testicular transcripts respond to FSH but not to sex steroids in the testes of zebrafish [95] and rainbow trout [96]. Most of these genes belong to cellular pathways known to regulate cell proliferation and differentiation, such as insulin-like growth factor (Igf3), Insl3, transforming-growth factor members (Tgf-β), Wnt, Notch, and Hedgehog signaling [95]. Since Sertoli cells express FSH receptor [88,89,90], it is likely that Sertoli cells act as a paracrine relay station for FSH signal in the testes [15]. 

Most of the accumulated knowledge regarding the role of growth factors in spermatogonial self-renewal vs. differentiation was derived from zebrafish studies (Figure 6) [20,21,89,95,97,98]. The Tgf-β Amh, expressed in Sertoli cells [97,98,99], has been shown to exert a role in adult teleost gonad development in both males and females, particularly at germ cell early stages [100]. In zebrafish and Japanese eels, Amh is characterized as a “spermatogenesis-preventing substance” [97,101]. Studies of zebrafish have demonstrated that Amh counteracts gonadotropin-induced effects on Leydig cell steroidogenesis [97], inhibits Sertoli cell pro-differentiation Igf3, and stimulates inhibitory factors such as inhibin-α and prostaglandin E2 (Figure 6) [98]. These data clearly demonstrate that Amh inhibits spermatogonium differentiation through modulation of growth factor production and suppression of Leydig cell function [97,98], maintaining spermatogonia in an undifferentiated state (Figure 6). 

The other well-known studied growth factor, Igf3, has been shown to be expressed exclusively in the gonad tissue in several teleost species [21]. In the testes, Igf3 has been detected in Sertoli cells [20,102], as well as in undifferentiated and differentiated spermatogonia, spermatocytes, and spermatids of tilapia [102]. There is evidence that FSH stimulates Igf3, which, in turn, promotes spermatogonial proliferation and differentiation in zebrafish testes (Figure 6) [20]. A recent study has shown that FSH-stimulated Igf3 release activates β-cateninc signaling in type A spermatogonia to stimulate their differentiation (Figure 6) [103]. 

Accumulated evidence in zebrafish demonstrates that FSH stimulates spermatogonial differentiation by down-regulating Amh and activating β-catenin signaling via Igf3. In addition, FSH promotes spermatogonial self-renewal through Wnt5a and the non-canonical Wnt pathways [21]. This balanced regulation could counteract depletion of A_und_ spermatogonia while promoting spermatogonial differentiation (Figure 6).

## 2. Isolation and Enrichment of Germ Cells in Fish 

### 2.1. Enzymatic Digestion

Isolation of germ cells by enzymatic digestion has been widely applied in both mammalian and teleost species [104,105,106,107]. The technique involves decapsulating and mechanically mincing the gonads, followed by dissociation via incubation with enzymes. Combinations of collagenase, trypsin-EDTA, dispase, and DNase I are commonly used. 

In fish, trypsin and collagenase are the most common enzymes employed in gonad dissociation [108,109,110,111,112]. Okutsu et al. [51] reported use of 0.5% trypsin to digest testes of rainbow trout, and the obtained spermatogonia produced viable offspring when transplanted into masu salmon *Oncorhynchus masou* larvae. Lacerda et al. [48,79] dissociated testes of tilapia using collagenase, trypsin, and DNAse I. For ovaries, enzymatic dissociation has been performed with collagenase in rainbow trout [113] and zebrafish [114]. 

Variables to be considered for enzymatic dissociation include species, sex, enzyme selection and concentration [115], and exposure time [111]. Insufficient digestion may cause low yield of single cells, while over-digestion may lead to cell damage. Disruption of epitopes from the cell membrane surface, which may impair germ cell function, has been observed during trypsin digestion [76,116,117]. Dispase is reported to be a relatively mild enzyme with which to minimize damaged effects on cells [118]. In sterlet, trypsin appeared to be more effective at dissociating germ cells from gonadal tissue when compared to collagenase [111]. More studies to address the effects of trypsin on fish germ cells are needed. 

### 2.2. Germ Cell Purification

Enrichment and purification are required to counteract gonad cell heterogeneity and low quantities of spermatogonia/oogonia. Cell isolation techniques are used to separate cell populations with respect to physiochemical and biochemical properties, including size, density, electrostatic characteristics, and differential expressions of specific cell surface markers [119]. Germ cell purification methods have been established to isolate homogenous populations using centrifugal elutriation [120], density gradient centrifugation [48,121,122], differential plating [123], FACS [124,125], MACS [76], and combinations of those methods [57,68]. 

#### 2.2.1. Density Gradient Centrifugation

Discontinuous density gradients, such as Percoll and Ficoll, to separate cell populations based on density, are commonly used for germ cell purification, and are appropriate for spermatogonia of several fish species [10,126]. This technique has been widely applied to enrich spermatogonia before culture or transplant in mice [48], sheep [127], primates [128], and humans [129]. 

In Nile-tilapia, Lacerda et al. utilized Percoll to isolate the most immature spermatogonia before transplantation [48,79]. In loach *Misgurnus anguillicaudatus* [121], 60% of type A and early type B spermatogonia were collected in the 30–36% Percoll fractions, while late type B spermatogonia, spermatocytes, and spermatids were distributed in the 40% fraction. Wong et al. found the majority of zebrafish spermatogonia in the 25% and 30% Percoll fractions [122]. Percoll is currently the most commonly used method to isolate and purify fish spermatogonia [112,130,131]. However, the technique has low resolution capacity, since it may not distinguish spermatogonia from Sertoli cells, peritubular myoid cells, and Leydig cells. This contamination has been reported in rainbow trout [132], sterlets [111], channel catfish *Ictalurus punctatus,* and blue catfish *Ictalurus furcatus* [62]. 

In contrast to what is seen during spermatogenesis, germ cells increase in size during oogenesis, with oogonia being slightly smaller than early-stage primary oocytes [133]. The ovary also contains a larger quantity of somatic cells. Since oogonia exhibit an intermediate size among ovarian germ cells, their isolation is more challenging compared to that of spermatogonia. Wong et al. [114] detected germ cell distribution in Percoll fractions in transgenic zebrafish (*vasa:DsRed2-vasa*). The majority of ovarian germ cells were collected in the 25–35% Percoll interface, which provided approximately 20-fold enrichment of the initial cell suspension. The enriched cells were able to colonize the transplant recipient gonad and produce offspring. Wong et al. [122,134] also showed that Percoll-enriched oogonia were able to proliferate in vitro. 

#### 2.2.2. Differential Plating

Differential plating is a classic method in cell culture based on the adhesive properties of the cells. Unlike somatic cells, SSCs will adhere to laminin or feeder cells rather than directly to a culture plate in in vitro conditions [135]. Spermatogonial stem cells are more likely to adhere to a feeder layer after somatic cell attachment to culture plates. Based on this attachment, differential plating can eliminate the somatic cells. However, it is possible to lose a substantial quantity of adherent germ cells if they are harvested at an inappropriate time. Currently, differential plating is usually combined with other separation methods. 

As an example in mammals, based on the velocity of sedimentation and differential attachment, cell populations containing up to 70% swine SSCs were obtained, 80% of which were viable [136]. Differential plating applied in cattle can eliminate most of the somatic cells in culture [137]. The purity of rainbow trout spermatogonia can reach >95% using differential plating [110,115]. Lacerda et al. [68,79] combined Percoll and differential plating to enrich tilapia spermatogonia. Although differential plating is an effective method of obtaining high purity in germ cells without causing damage, enrichment usually takes 5–7 days, and cell properties might change during the in vitro culture period [110]. 

#### 2.2.3. Flow Cytometric and Magnetically-Activated Cell-Sorting

Flow cytometry, as well as FACS, has been employed for cell sorting for more than four decades [138]. Fluorescence-activated cell sorting is a rapid and quantitative technique with which to examine individual cells using a range of fluorescent and light scattering signals related to cell size, shape, granularity, surface, intracellular protein, and gene expression. When a heterogeneous group of cells is passed through a laser system, characteristics such as morphology, viability, and surface markers indicated by light scattering and fluorescence are captured and analyzed, and cells with defined signals can be collected [139,140,141]. For some transgenic fish, target cells can be sorted by FACS according to their self-fluorescence properties. In some endangered and cultured species in which transgenic techniques are not suitable or available, high resolution immunoaffinity techniques are generally conducted, based on the number and type of molecules present on the cell surface that can be targeted by specific monoclonal antibodies conjugated with florescence or magnetic microbeads. 

In mammals, FACS has been used for sorting SSCs from mouse testicular cell suspension [142]. Spermatogonial germ cells are highly enriched with characteristics such as α6-integrin positivity, c-kit expression, low side scatter, and negative or low av-integrin expression [142]. Other SSC surface markers that have been identified and used to sort and enrich mammalian SSCs include CD9, ITGB1, ITGA6, GFRA1, EPCAM, NCAM1, THY1, CDH1, SSEA-4, and MCAM [143,144,145,146,147]. Recently, FACS has been employed for separating spermatogonia from cancer cells in monkey testis [148], indicating a potential application in cell therapy. 

In tilapia, cell populations differentially stained with propidium iodide and carboxyfluorescein succinimidyl ester were identified and quantified by FACS [149]. However, there are few studies of germ stem cell surface markers. Nagasawa and colleagues [80,81,150] identified lymphocyte antigen 75 (Ly75/CD205) as a surface marker of primordial germ cells and mitotic germ cells in rainbow trout and Pacific bluefin tuna *Thunnus orientalis*. However, there is still no evidence that Ly75/CD205 is capable of labeling and fractionating live type A spermatogonia. To isolate and enrich type A spermatogonia, Yano et al. used pvasa-GFP transgenic rainbow trout and sorted cells based on diameter and green fluorescent protein (GFP) intensity [54,151]. Further study has shown that physiochemical characteristics such as high forward scatter and low side scatter could be used for sorting type A spermatogonia from GFP-transgenic rainbow trout [124]. These studies showed the possibility that spermatogonia can be dramatically enriched in a population of large cells with a simple intracellular structure. Subsequently, Ichida et al. [125] established a method of purifying type A spermatogonia of immature, maturing, and spermiogenic testes of Pacific bluefin tuna according to light scattering properties using FCM. In this study, spermatogonia were enriched 15-fold compared to the unsorted cell fraction. These findings indicate that light scattering properties are applicable to enriching type A spermatogonia without cell-labeling systems such as transgenes and cell surface antibodies. Monoclonal antibodies (numbers 172 and 189) have been produced by inoculating Pacific bluefin tuna-enriched live type A spermatogonia into mice, and screened using cell-based enzyme-linked immunosorbent assay (ELISA), immunocytochemistry, FCM, and immunohistochemistry [77]. These antibodies were able to recognize cell surface antigens of Pacific bluefin tuna type A spermatogonia and be used to identify live spermatogonia in a recipient following transplanattion when conjugated with fluorescent dye [77]. This represents an important advantage in applications to commercially valuable or endangered species that lack transgenic strains or specific molecular markers for identifying cell lineages.

Flow cytometry is an automated, multiparametric, and sophisticated sorting tool requiring skill and costly equipment, and it is not affordable for many laboratories. Moreover, it can require cell labelling with fluorescent antibodies, which might alter cells and their functions.

In contrast, MACS does not require a flow cytometer and can be performed in a short period of time through relatively simple methodology. It uses magnetic beads that bind specifically to target cells with mAbs [152]. The dissociated testicular or ovarian cell suspension is incubated with magnetic nanoparticles that are directly or indirectly conjugated with mAbs against a particular surface antigen. Once bound to the intended target, beads are subjected to magnetic forces that allow immobilization of the bound cell type and concurrent separation from other components in the suspension. Additional washing and elution steps complete the purification cycle, resulting in an enriched preparation of a specific cell type [152]. This technique can also be used for negative selection to eliminate undesired cells. Compared with methods requiring FCM, MACS is a simple technique and requires no special equipment except the magnetic beads and a magnet stand. The technique is effective for large quantities of cells and more rapid than FCM at collecting a high number of cells [113]. 

Magnetically-activated cell sorting is commonly used in mammalian germ cell research [153,154,155,156], with several surface markers employed to differentially sort spermatogonia. The c-Kit protein has been used to separate types of spermatogonia in hamsters, mice, and monkeys [157]; GFRα1 [158], α6-integrin [159,160], CD9 [161,162], and Thy-1 [153,163,164] have been used to isolate SSCs. Most of these surface markers have also been applied to sorting fish spermatogonia. Panda et al. [57] enriched highly pure type A spermatogonia from carp testis using Thy1.2 (CD90.2) antibody from mouse. The same marker has been employed in channel catfish *I. punctatus* and blue catfish *I. furcatus* spermatogonia [9]. 

It is believed that cell surface markers should exhibit low species-specificity, but mammalian antibodies are not always suitable for work with fish spermatogonia. For this reason, Yoshizaki and colleagues produced cell surface mAbs by inoculating enriched and live type A spermatogonia from vasa::GFP rainbow trout into mice [75]. Using these antibodies, it was possible to isolate undifferentiated germ cells from vasa::GFP rainbow trout with MACS [75]. The MACS-enriched cells showed a strong GFP signal and significantly higher transplantability than the unsorted cells, especially the ovarian cells [76]. However, like in FACS, antibodies bound to the cell surface may inhibit some cell functions, since cell surface proteins or sugar chains could be masked by the antibodies [77]. In addition, the surface markers/antibodies developed for MACS may not be effective for all species. It is also lower resolution than FACS, since some antibodies identified by FACS are not recognized by MACS [147]. Therefore, more studies regarding antibody identification in fish are required.

### 2.3. Future Trends

Recently, microbeads and nanobeads have been used to isolate human stem cells and cancer cells [9,165] and provide new possibilities for sorting fish germ stem cells. Microbeads and nanobeads enable rapid binding and short labeling procedures [9,165]. Due to their small sizes, these particles do not saturate cell epitopes and do not interfere with downstream applications. Other novel separation methods in the stem cell field, such as aptamer-based separation [166] and affinity chromatography, can be considered potential tools for sorting germ stem cells.

## 3. Germ Cell Cultures

### 3.1. Serum in Germ Cell Cultures

Serum is used to supply essential nutrients for in vitro cell growth [167]. Serum is a mixture containing plasma proteins, polypeptides, growth factors, hormones, and binding proteins along with contact and extension factors that protect cells from damage when they adhere to culture plates [168]. It may also contain unknown components that affect cell growth. The most commonly used serum in cell culture is fetal bovine serum (FBS). Commercial media, such as Leibovitz’s L-15 medium and Dulbecco’s modified eagle’s medium, supplemented with FBS, have become widespread in vertebrate cell culture. In early fish germ cell culture, FBS was used to support germ cell proliferation [167]. There is evidence that rainbow trout type A spermatogonia proliferate in culture conditions that include a high FBS concentration in the medium, but propagation eventually ceases due to the overgrowth of somatic cells [115]. Because rainbow trout type A spermatogonia have an extremely slow cell cycle, somatic cells gradually occupy space and nutrition in the culture, especially at higher serum concentrations [115]. In mice, FBS concentrations of 0.3% to 2% in media allow initial somatic cell attachment and proliferation on flasks and subsequent SSC attachment to somatic cells with colony formation after 5 to 7 days of culture [115]. However, when a higher concentration (5–15%) of serum was used, SSCs propagated and formed colonies, but eventually ceased growth and detached, due to the extensive growth of fibroblasts [169]. Thus, high serum concentrations may stimulate germ cell proliferation, while simultaneously causing extensive growth of somatic cells, inhibiting continued germ cell proliferation. Reports have shown that germ cells cultured with FBS were not able to achieve long-term proliferation and maintenance of original characteristics in zebrafish testicular cell culture [105,170]. McClusky [171] reported that FBS simultaneously stimulated both in vitro apoptosis and [3H]thymidine incorporation in immature spermatogonia in a concentration-dependent manner. Fetal bovine serum contains a wide range of both inhibitory and stimulatory factors [172,173]. It is possible that germ cells and/or their associated Sertoli cells are responsive to both inhibitory and stimulatory signals, resulting conflicting signals, and ultimately, cell death. 

Bone morphogenic protein 4 (BMP-4), a stimulatory factor present in FBS [174], can induce differentiation in cultures of zebrafish [175] and mouse spermatogonia [87,176,177]. To overcome this, substitutes for serum are being explored. A serum-free culture of mouse SSCs was first developed by Kutoba [163]. Mouse SSCs could maintain proliferation more than six months on minimum essential medium-alpha supplemented with 0.2% bovine serum albumin (BSA). Recently, Aoshima et al. demonstrated that “knockout serum replacement” medium maintained long-term growth of SSCs in the absence of BSA and serum [178]. In fish, to suppress the overgrowth of testicular somatic cells, Shikina and Yoshizaki replaced FBS with soluble factors, such as BSA, adenosine, and salmonid serum [110]. The new culture extended the duration of type A spermatogonia culture, maintaining their original morphology and GFP intensity similarly to that in *vas::GFP* trout spermatogonia, without overgrowth of somatic cells [110]. In zebrafish, spermatogonia were reported to show effective propagation for up to three months in medium with 3% FBS [179]. Wong et al. established a female germline stem cell (FGSC) culture based on a serum-supplemented, proprietary, StemPro-34-based (Gibco) medium, which contained the original StemPro-34 supplement plus 16 individual compounds along with basic nutrients, including 1% FBS [122]. The proliferation of FGSCs continued for more than 6 weeks in vitro with unchanged germ cell markers, and cells generated normal offspring after transplantation. Multiple germ cell cultures with decreased, or no, FBS for long-term maintenance have been reported [60,122,134]. Research should focus on development of a well-defined medium for mixed cell populations that inhibits growth of nontarget cells while stimulating proliferation of target cells.

### 3.2. Feeder Cells and Growth Factors

Fish germ stem cell survival, self-renewal, and differentiation are regulated by a combination of intrinsic and extrinsic factors [10,97,180]. Germ cells are capable of autonomous control of the differentiation pattern, and somatic cells, especially Sertoli cells, support germ cell development and provide growth factors that modulate germ cell proliferation and fate. The growth factors released by Sertoli cells are required for germ stem cell proliferation and differentiation [181], as well as initiation of meiosis [182]. 

In 1994, Loir [183] cultured rainbow trout spermatogonia with L-15 supplemented with Ultroser^TM^ serum substitute medium and 5% rainbow trout serum. In the presence of Sertoli cells, spermatogonia survived for two weeks [183]. A co-culture system of germ cells and somatic cells was established for the Japanese eel in 1996 [184], in which the aggregated cells undergo spermatogenesis in the presence of 11-Ketotestosterone. In 2002, using testis-derived tumor-like cells, designated ZtA6—expressing *sox9a* and phagocytotic activity (Sertoli cell features)—as feeder cells, Sakai et al. reported that male zebrafish germ cells went through mitosis and meiosis and developed into functional spermatozoa in vitro [105]. Prolongation of vasa expression in the culture suggested that ZtA6 cells promoted survival and propagation of spermatogonia [105]. In addition, ZtA6 cells contributed to the attachment and survival of spermatogonia at the beginning of culturing [105]. The formation of flagellated spermatozoa was observed on day nine of culture, and clusters of spermatogonia disappeared on day 20. In this culture system, all types of germ cells, including type A and B spermatogonia and primary and secondary spermatocytes were cultured, precluding accurate identification of the type of germ cell stimulated by the feeder cells. Feeder cells were heterogeneous, suggesting that somatic cells other than Sertoli cells could be responsible for germ cell proliferation and spermatogenesis. Kurita and Sakai further isolated and established two Sertoli cell lines (ZtA6-2 and ZtA6-12) from testis-derived tumor-like ZtA6 cells [185]. The ZtA6-2 cell line was able to support germ cell growth, and ZtA6-12 promoted germ cell differentiation into spermatozoa [185]. The feeder cells were successfully employed in production of spermatozoa from cultured spermatogonia [170]. The transgenic spermatozoa were differentiated from premeiotic germ cells which were transfected with a pseudotype retrovirus in vitro. In medaka, primary culture of testicular cells has shown formation of spermatozoa after 24 h [186]. The culture generated both adherent and suspended cells. Proliferating cells and differentiation of spermatocytes into spermatids or spermatozoa were observed mainly among the suspended cells. Interestingly, the primary cells showed a dynamic equilibrium between adherent and suspended cells and could not be separated into their respective cell populations. This study suggested that medaka germ cells are more likely to spontaneously initiate spermatogenesis than to remain undifferentiated [186].

A normal spermatogonial cell line (SG3) derived from a mature medaka testis was established in 2004 [108]. After 140 passages, the SG3 cell line remained stable with respect to proliferation, karyotype, and phenotype when cultured with basic fibroblast growth factor (bFGF) and medaka embryo extract. The SG3 cell line expressed spermatogonial stem cell genes rather than Sertoli cell markers. In this culture system, spermatogonial cells proliferated in the absence of feeder cells, but ceased rapid growth upon depletion of growth factors in the culture medium. The SG3 cell line was observed to undergo meiosis and spermiogenesis to generate motile spermatozoa in vitro [108]. Similar to a previous study [186], spermatogenesis occurred in suspended aggregated germ cells. Spermatogenesis was automatically induced by high cell confluence without subculture. It has been suggested that bFGF may inhibit differentiation or promote self-renewal in medaka [108], indicating that, although spermatogonia exhibit potential for continuous proliferation, their long-term culturing depends strongly on culture conditions. 

In mammalian cultures, various feeder layers have been tested. The effects of the Sandos inbred line, mouse embryo-derived, thioguanine-resistant, ouabain resistant cells; mouse embryonic fibroblasts; and mouse testicular stromal cells on SSC proliferation and maintenance differ between species [87,187,188]. All these substances interfere with stability of feeder cells and make the culture process more difficult to control. Mouse SSCs have been successfully cultivated without feeder cells on a laminin-coated plate in the presence of GDNF [169]. 

As germ-cell xenotransplantation has been established for rainbow trout [55,189,190], the high demand for germ stem cells requires an in vitro culture system to propagate and maintain spermatogonia for long periods before transplant. Using immature rainbow trout, Shikina and Yoshizaki reported optimized spermatogonium survival and proliferation in type A undifferentiated spermatogonia with culture in L-15 supplemented with 10% FBS [115]. Spermatogonial proliferation was maintained for more than one month in vitro by addition of insulin, trout embryonic extract, and bFGF to the culture medium [115]. However, the overgrowth of testicular somatic cells restricted proliferation of spermatogonia after one month of culture. The cultured spermatogonia colonized and proliferated in the recipient gonads following transplantation, although at a lower rate than observed for fresh disassociated cells [115]. 

To further suppress the overgrowth of testicular somatic cells and enhance the survival, proliferation, and transplantability of spermatogonia, Shikina and Yoshizaki replaced FBS with adenosine and salmonid serum [110]. These conditions enhanced the transplantability of cultured spermatogonia, indicating that adenosine and salmonid serum were able to stimulate SSC survival and self-renewal in vitro. 

The development of culture media has increasingly been focused on improving germ cell expansion while maintaining “stemness.” Initial studies expressed uncertainty as to whether germ cells in clumps grow during in vitro culturing [105,108,110]. Several studies observed spermatogenesis after clump formation, suggesting that differentiation may have occurred [108,186]. However, SSC proliferation may also take the form of clumps. In *L. rohita*, enriched SSCs sorted by MACS with Thy 1+ were capable of proliferation and remaining undifferentiated for more than 60 days [57]. In this feeder-free culture, cells proliferated to form colonies with unique morphological compactness features. The induction of proliferation was not observed in the presence of either mouse or human recombinant GDNF in the culture medium [57]. In zebrafish, most clumps of spermatogonia were positive for BrdU, indicating active proliferation [179]. Lacerda [68] showed that Nile tilapia spermatogonia remained undifferentiated and able to proliferate for at least one month of culture. When purified spermatogonia were kept at high confluence without subculture, large colonies expressing *vasa*, *sox2*, and the *gfra1* were generated, without expression of meiotic marker (*dmc1*). Thus, it could be assumed that clumps were likely derived from a few aggregated SSCs and expanded during proliferation [68].

Spontaneous germ cell differentiation has been observed, and probably can be attributed to cell-cell communication [2,3,4]. Fish embryo stem cells and spermatogonia have been demonstrated to differentiate in vitro via enhanced cell–cell interactions [191]. Nevertheless, it is possible that, in contrast to in vivo, the surface properties of the cells may change in vitro and the adhesiveness reflect specific developmental and/or cell cycle phases.

In 2012, Kawasaki et al. showed that zebrafish spermatogonia proliferate continually for longer than one month in a culture medium supplemented with a cocktail of recombinant mammalian growth factors, including GDNF, IGF-1, and bFGF. The cultured spermatogonia produced functional spermatozoa after transplantation [179]. In contrast to previous studies in which GDNF showed no effect on germ cell proliferation [57], in the zebrafish culture, Kawasaki and collaborators showed that recombinant human GDNF enhanced spermatogonial proliferation [179]. However, compared to mouse SSC culture, a 5 to 10-fold concentration of GDNF was needed to promote proliferation of zebrafish spermatogonia [163,192]. In 2013, Wong et al. further studied the role of homologous growth factors [122] in homogeneous cultures of zebrafish FGSCs. The FGSCs remained undifferentiated for more than six weeks in vitro and showed high transplantability [122]. The FGSC proliferation was enhanced when cultured on ovarian-somatic feeder cells (OFCs) engineered to express zebrafish leukemia inhibitory factor (LIF), bFGF, and GDNF in a serum-free culture [122]. These conditions were also applied to a zebrafish SSC culture [134,175] with the opposite effect observed: the number and proliferation of spermatogonia decreased, and differentiation was stimulated after three weeks [175]. The OFCs used in this culture were assumed to express Bmp and induce spermatogonium differentiation, since the expression of Bmp was increased in the latter part of the culture period. When blocking Bmp signaling with the specific inhibitor dorsomorphin, spermatogonium growth could be prolonged to 6 weeks. Thus, the effect of Bmp was shown to differ in culture of zebrafish male and female germline stem cells. 

In culture of dogfish spermatogonia, GDNF has been shown to induce spermatogonium proliferation by reducing apoptosis, and cells remained pluripotent and capable of self-renewing for more than five months [60]. Similarly, recombinant medaka GDNFa/b enhanced the proliferative activity of SG3 cells, which retained the spermatogonial gene expression pattern and showed alkaline phosphatase activity [193]. There is also evidence that a high concentration of recombinant human GDNF and LIF in culture medium was responsible for maintaining spermatogonia of sturgeon in vitro [194]. Satoh et al. examined the in vitro effects of baculovirus-produced recombinant medaka LIF proteins on medaka spermatogenesis [195], and found that LIF protein in the culture medium and in co-culture with LIF-overexpressing medaka testicular somatic cells promoted spermatogonial proliferation [195]. The influences of physiological and biochemical factors on germ cell culture have been investigated [196,197]. Kawasaki et al. found high oxygen (20%) concentration to stimulate spermatogenesis, while heparin contributed to propagation of spermatogonia [196]. 

In summary, supplementation of culture media with appropriate soluble growth factors is essential for germ cell survival and proliferation for long-term incubation. However, in co-culture with somatic cells, germ cells are likely to attach to somatic cells, form clumps, expand in number, and initiate spermatogenesis. This indicates that germ cell in vitro activity may occur partly through intercellular junctions and not only through paracrine signals from Sertoli cells. The germ cell attaches to feeder cells prior to initiating proliferation. Some studies have demonstrated that Sertoli cell-germ cell and germ cell-germ cell adhesive and gap junctions are involved in Sertoli cell functions to support germ cell development within the cyst [198,199]. Thus, the role of intercellular junctions on germ cell development, and whether there is synergism/antagonism between soluble growth factors and the cell junctions need to be investigated. 

### 3.3. Organ Cultures

In general, cell culture systems are not suitable for evaluating complex signal pathways among cells and the extracellular matrix [200]. In a germ cell culture, it is difficult to control the re-association of somatic and germ cells, and feeder cell layers may be required. In order to study gametogenesis and its endocrine and paracrine control ex vivo, an effective organ culture system that allows germ cells to maintain their spatial arrangement and their normal cellular and microenvironmental composition when grown in vitro must be established. Organ culture must also maintain the integrity of the complex regulation existing in the gonads. 

The effort to develop a testis tissue culture system began in the 1960s in mammals [201]. Trowell [202] developed an organ culture system, and Steinberger et al. [203] adapted it to rats. It is a method in which tissue fragments are placed on a thin layer of agarose covering a perforated metal grid (Figure 7A). Calf serum (10%) was added to maintain the viability of the seminiferous epithelium [203]. This agarose gel organ culture became a reliable standard method to study in vitro spermatogenesis in mammalian species including rabbits and humans [204,205,206].

Organ culturing for spermatogenesis was reported in immature Japanese eels in the 1990s [207]. Similarly, to techniques use in mammals, freshly minced eel testes were placed on floats of elder pith covered with a nitrocellulose membrane in tissue culture dishes (Figure 7B). The basal culture medium consisted of L-15 supplemented with BSA, a cocktail of amino acids, and other substances necessary to maintain cell activity. This culture system has been employed in analyzing effects of several steroid hormones on eel spermatogenesis, and, with some modifications, in studies of Japanese Huchen Hucho perryi testes and ovaries [210,211]. In 2009, a zebrafish testes culture system was developed, based on the Japanese eel model [208]: Following disinfection, testes were placed on a 0.25 cm^2^ nitrocellulose membrane in a 750 μm agarose cylinder and kept in 1 mL medium in each of 24-well flat-bottom plates (Figure 7C). The basal culture medium was composed of L-15 supplemented with HEPES, BSA, retinoic acid, and antibiotics [208]. The agarose cylinder was prepared using 48-well plates as molds [208]. The zebrafish testis culture model has been widely used to study endocrine and paracrine regulation of zebrafish spermatogenesis [20,22,89,212].

A Nile tilapia testis culture model similar to the eel system was developed to study sex differentiation in vitro [213]. A testis tissue culture system has also been established to examine effects of FSH and LH on gene expression in rainbow trout testis [96,214]. A novel culture system capable of effectively replicating the microcirculatory system of the mouse body was developed in 2016 [215]. The device consisted of a chamber for tissue and a channel for medium flow. The testis tissue was separated from the flowing medium by a thin porous membrane. Nutrients and waste products were exchanged by molecular diffusion as in vivo (Figure 7D). Using this microfluidic culture method, mouse testis tissue could be maintained six months in vitro and could undergo spermatogenesis [215]. With the addition of appropriate hormones to the culture medium, the system can mimic in vivo conditions. In 2018, a simpler, user-friendly system [201] using hydrostatic pressure of the medium in a reserve tank, along with a resistance circuit, was developed to control the flow rate without the use of a pump. The pumpless device can create culture conditions for testis tissue in a tissue chamber (Figure 7E). These microfluidic devices have successfully created a novel organ culture system that differs from the classic agarose gel system and has the potential to revolutionize organ culture.

The current organ culture systems described for fish can also support culture conditions suitable for ex vivo study of testis function. Organ culture can maintain the integrity of gonadal processes and, hence, is a powerful tool to evaluate the effects of a wide range of substances involved in fish spermatogenesis, among them pituitary hormones, steroids, and growth factors, as well as suspected endocrine disruptors. 

### 3.4. Three-Dimensional Cultures

The ability to view the gonad as a three-dimensional structure, as it is in vivo, could be valuable for study of testis development and spermatogenesis. Three-dimensional (3D) models currently represent an optimal tool to provide a tissue-like context for cell culture. In 3D cell cultures, the communication between cells and the scaffold is regulated by the material characteristics and scaffold properties [216]. In order to enhance cell adhesion, proliferation, and activation, materials employed in the fabrication of scaffolds must possess characteristics involved in intrinsic biocompatibility along with the appropriate chemistry to induce molecular recognition from cells. Three-dimensional culture systems were initially established for clonogenic assays to explore the complex mechanisms of multipotent hematopoietic cell proliferation and differentiation [217]. Later they were widely applied to stem cell studies, including exploring the impact of different scaffolds on mammalian SSC culture. Lee et al. first reported the reconstruction of the tubular structure of rat testis using collagen gel and Matrigel (BD Biosciences) [218], obtaining germ cell differentiation with a mixture of germ cells and testicular somatic cells cultured in a 3D system. In humans, spermatocytes have been induced to differentiate into presumptive spermatids in a collagen gel matrix culture [219]. A novel method using a soft agar culture system has become a common method for culture of stem cells and has also been applied on mouse SSCs [220,221,222,223]. Another 3D culture model using a bicameral chamber was developed by Legendre [224]. In fish, there are few studies of 3D culture, but fish collagen (FC) (Figure 8) is considered an ideal scaffold material due to its biocompatibility, low immunogenicity, low cytotoxic effects, high cell viability, and high biodegradability [225]. Studies have evaluated the FC of several species, including bester (beluga *huso huso* × sterlet) and tilapia, as 3D culture scaffold materials [225,226,227,228,229]. To best of our knowledge, a 3D culture of fish germ cells has not been reported. This system is considered a novel technique for study of fish gametogenesis, with methods and protocols to be standardized. 

A three-dimensional culture creates a complete microenvironment resembling the in vivo conditions. Although such systems have been shown to successfully induce spermatogenesis, conditions differ from those *in vivo*, and comparisons may be inaccurate. Using an appropriate scaffold, 3D culture can avoid the ischemia that develops in long-term organ tissue culture and mimic the structure of germ cells within the cyst.

## 4. Conclusions

This review summarizes the most relevant aspects of teleost germ cell biology, including spermatogonium characterization, and novel techniques for isolation and culture. Combined with such procedures as cryopreservation and transplantation, stem germ cell propagation can be a powerful tool for research, aquaculture, and conservation of endangered species (Figure 9). Stem germ cell isolation, enrichment, and culture conditions to support their proliferation without differentiation will help to overcome the limitations of low germ cell numbers for cryopreservation and transplantation (Figure 9). Germ cell culturing is valuable for studying the mechanisms underlying mitotic proliferation, differentiation of stem cells, meiosis, and stem cell regulation *in vitro*. Of particular interest is the potential to combine germ cell culture with gene editing techniques such as CRISPR/Cas for germ-line transmission by cell transplantation, nuclear transfer, and/or in vitro spermatozoon production for artificial insemination. Knowledge in this area has evolved rapidly, and has revealed novel and promising approaches to germ stem cell manipulation.

## Figures and Tables

**Figure 1 biomolecules-10-00644-f001:**
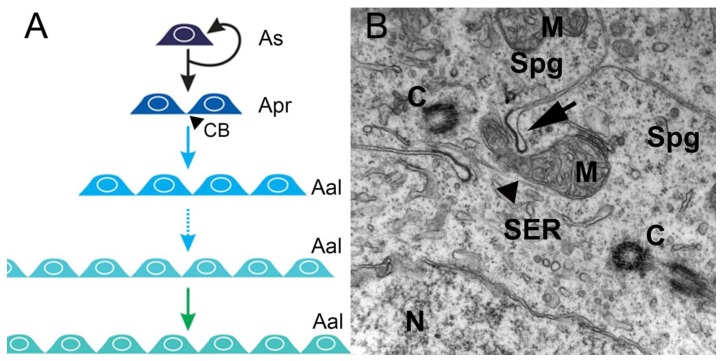
Mitotic/spermatogonial phase. (**A**) Mitosis produces spermatogonium clones. In rodents, type A single spermatogonia (As) harbor the spermatogonial stem cell population, which can either self-renew or generate two interconnected cells named type A-paired spermatogonia (Apr). The A-paired spermatogonia are interconnected by cytoplasmic bridge (CB) as a consequence of incomplete cytokinesis during cell division. Amplifying divisions beyond the A-paired also do not complete cytokinesis and continue to generate longer syncytial chains, termed A-aligned spermatogonia (Aal). (**B**). An electron micrograph showing a cytoplasmic bridge (arrow) connecting daughter cells resulting from spermatogonium division. The mitochondria (M), nucleus (N), smooth endoplasmic reticulum (SER), centrioles (C), and microtubules (arrowhead) are depicted in the cytoplasms of two interconnected spermatogonia (Spg) (illustration and data: Nóbrega—unpublished observations).

**Figure 2 biomolecules-10-00644-f002:**
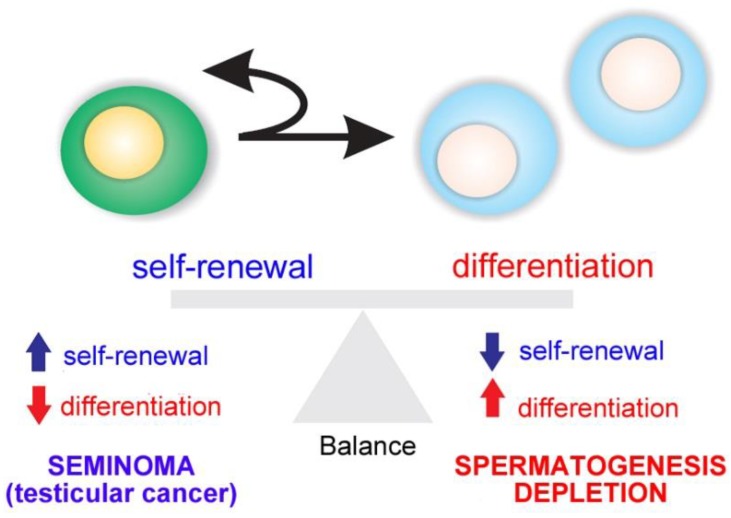
Balance between spermatogonial stem cell (SSC) self-renewal and differentiation. An imbalance results in testicular cancer or depletion of spermatogenesis.

**Figure 3 biomolecules-10-00644-f003:**
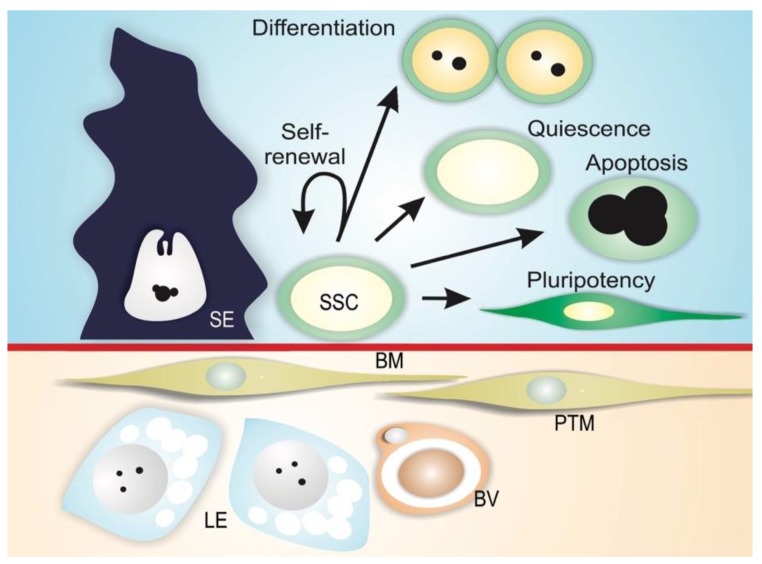
Mammalian SSC niche. Spermatogonial stem cells (SSC) reside along the basement membrane (BM) in proximity to interstitial Leydig cells (LE) and blood vessels (BV), and are in contact with Sertoli cells (SE). In this microenvironment, physical and paracrine interactions regulate SSC self-renewal, differentiation, quiescence, apoptosis, and the ability to transform into different cell types (pluripotency). Peritubular myoid cells (PTM) are also depicted in the figure.

**Figure 4 biomolecules-10-00644-f004:**
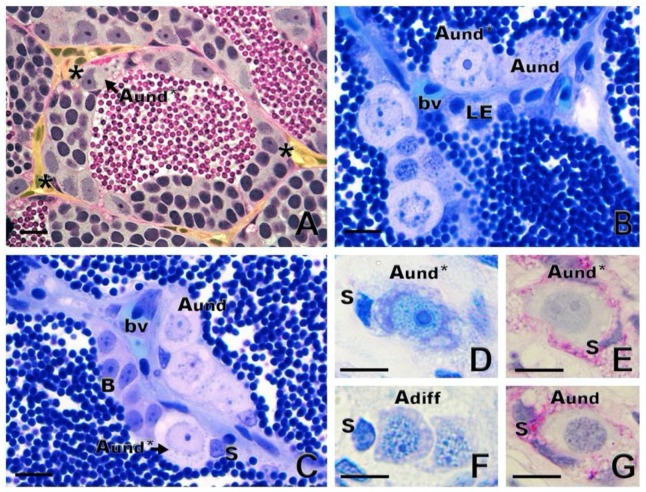
Spermatogonial niche and type A undifferentiated (A_und_) spermatogonia. (**A**) Asterisks show the most undifferentiated type A spermatogonia (A_und*_), preferentially located near the interstitial compartment (delimited in yellow) in zebrafish *Danio rerio*. (**B**,**C**) In common carp *Cyprinus carpio*, both A_und*_ and A_und_ are located near the interstitium, close to Leydig cells (LE) and blood vessels (bv). Sertoli cell (S) and type B spermatogonia are shown. (**D**–**G**): Generations of sterlet *Acipenser ruthenus* spermatogonia: A_und*_ and A_und_ and type A differentiated (A_diff_) spermatogonia. Staining: periodic acid–Schiff (**A**,**E**,**G**) and toluidine blue (**B**,**C**,**D**,**F**). Scale bar: 10 µm.

**Figure 5 biomolecules-10-00644-f005:**
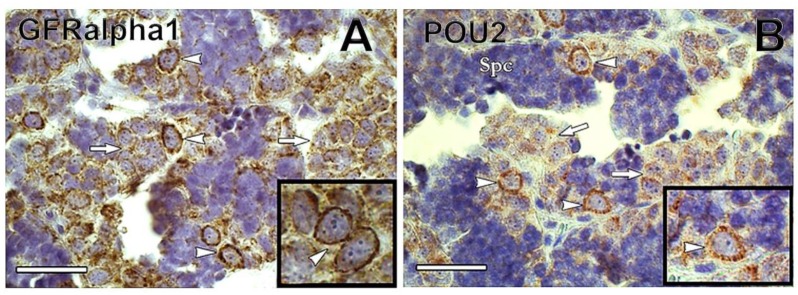
(**A**) Immunoreactivity of GFRα1 was found preferentially in type A undifferentiated spermatogonia (arrowheads) of common carp. In contrast, GFRα1 immunoreactivity decreased in differentiated spermatogonia (arrows). The inset shows GFRα1-positive undifferentiated spermatogonia (arrowheads). (**B**) POU2 was detected in common carp type A undifferentiated spermatogonia (arrowheads), whereas its immunoreactivity decreased in differentiated spermatogonia (arrows). Inset shows POU2-positive undifferentiated spermatogonia. Spc = spermatocyte. Scale bar: 25 µm.

**Figure 6 biomolecules-10-00644-f006:**
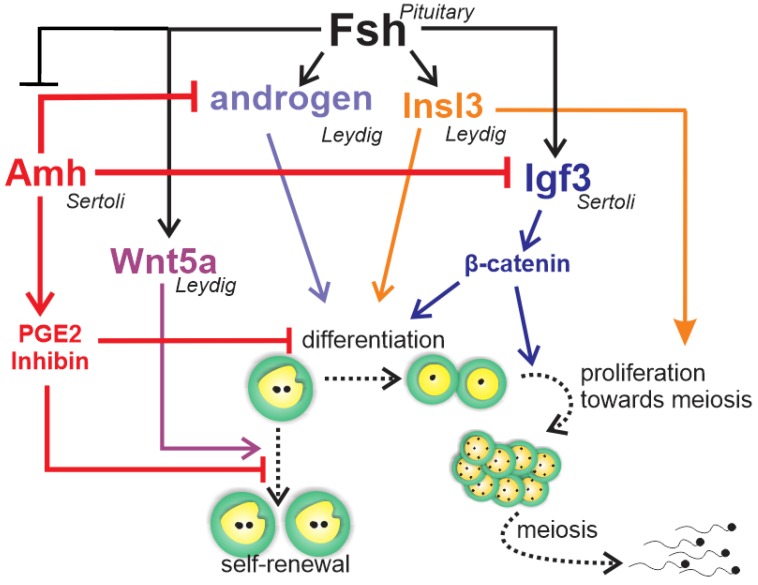
Schematic representation to summarize the regulation of zebrafish spermatogonial phase. Fsh exerts a central role in the zebrafish’s spermatogonial phase by triggering and balancing steroid and growth factor production in testicular somatic cells (Leydig and Sertoli cells). In Leydig cells, Fsh stimulates the production of androgens and Insl3, which are both involved in the spermatogonial differentiation and proliferation. Moreover, Fsh promotes spermatogonial self-renewal by increasing Wnt5a release by Leydig cells. In Sertoli cells, Fsh increases Igf3, which in turn promotes differentiation and proliferation by activating β-catenin signaling in the germ cells. Fsh also down-regulates Amh, a member of the TGF-β (transforming growth factor-beta) superfamily produced by Sertoli cells. Amh, through PGE2 or inhibin, exerts an inhibitory role on spermatogonial self-renewal and germ cell differentiation in the zebrafish testes. To sustain its inhibitory role, Amh also decreases androgen and Igf3 production in Leydig and Sertoli cell, respectively.

**Figure 7 biomolecules-10-00644-f007:**
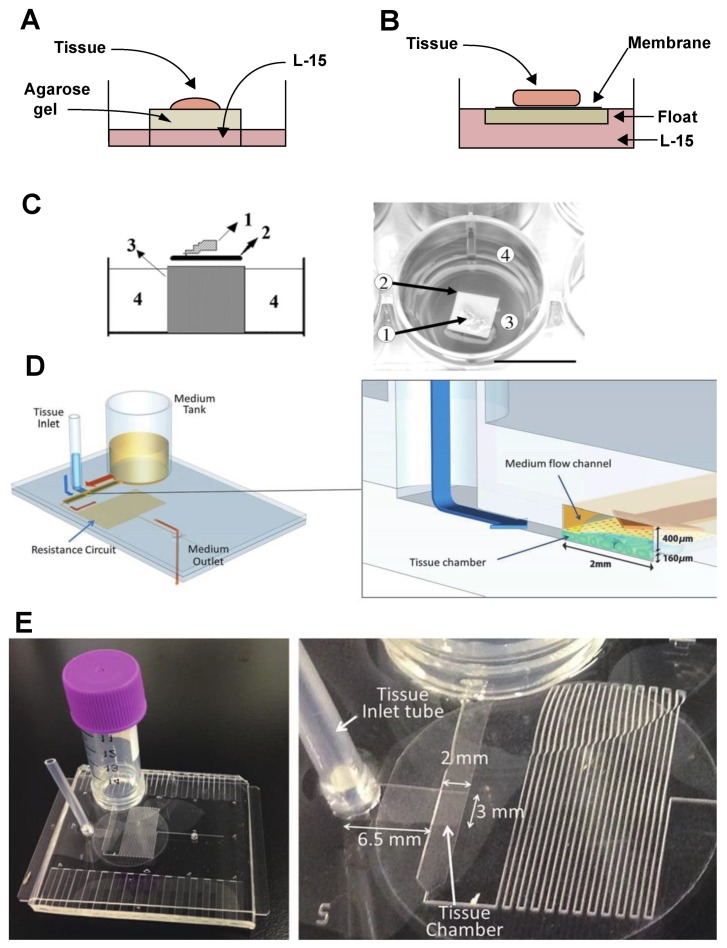
Organ culture systems. (**A**) The classical agarose gel model in mammals [203]; (**B**) the Japanese eel testis culture model [207]; (**C**) the zebrafish testis primary culture model [208] (1, testis tissue; 2, nitrocellulose membrane; 3, agarose cylinder; 4, medium (1 mL); scale bar: 1 cm); (**D**) Microfluid culture model [209]. (**E**) Pumpless microfluid culture [209].

**Figure 8 biomolecules-10-00644-f008:**
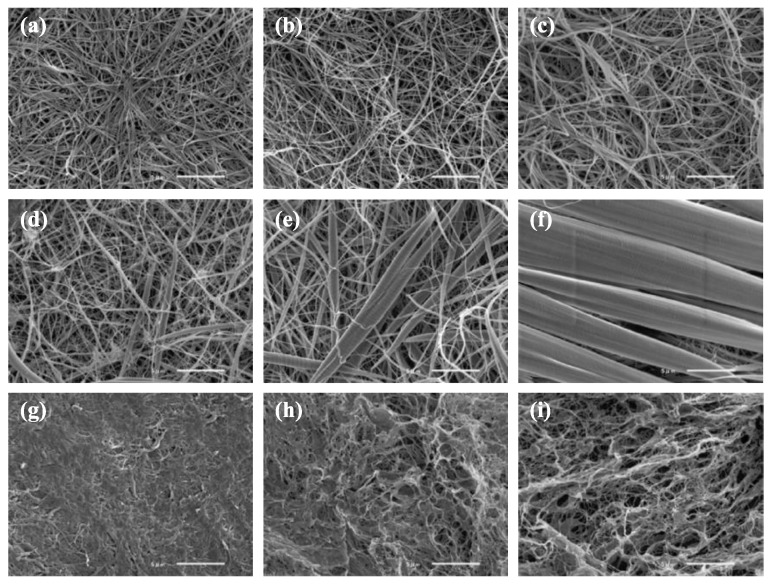
Scanning electron micrographs of skin collagen, swim bladder collagen, and porcine tendon collagen fibrils formed at 21 ± 1 °C for 24 h. (**a**–**c**) bester skin with NaCl concentrations of 0, 70, and 140 mM; (**d**–**f**) bester swim bladder with NaCl concentrations of 0, 70, and 140 mM; (**g**–**i**) porcine collagen with NaCl concentrations of 0, 70, and 140 mM. Scale bars, 5 μm [227].

**Figure 9 biomolecules-10-00644-f009:**
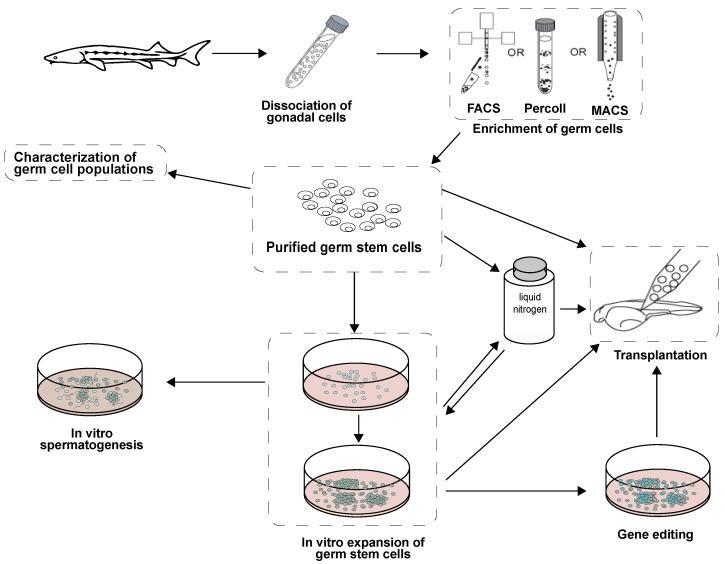
Potential applications of germ cell isolation and culture. Testes are removed from the fish body and dissociated, providing a cell suspension that can be sorted using fluorescence-activated cell sorting or centrifugation in a density gradient; e.g., in Percoll solution or by magnetic-activated cell sorting. The purified germ cell suspension can be characterized with respect to molecular characteristics and morphology. Purified cells may be injected into a recipient to generate germ-line chimera or amplified by in vitro culture. Cultured cells can be employed in gene editing or subjected to cryopreservation.

**Table 1 biomolecules-10-00644-t001:** Spermatogonial stem cell markers in fish.

Marker	Specification	Species	Reference
SGSA-1	Spermatogonia specific-antigen-1	Japanese eel	[78]
Notch1	Notch homolog protein 1	Rainbow trout	[79]
Pou5/2 (Oct-4)	POU domain, class 5/2	Medaka Common carp	[71] Nobrega et al. unpublished observations
Ly75 (CD205)	Lymphocyte antigen 75	Rainbow trout Pacific bluefin tuna	[80] [81]
PLZF	Promyelocytic leukemia zinc finger	Zebrafish Carpa rohu Dogfish Rainbow trout Catfish (several species)	[58] [57] [60] [61] [9,62,63]
GFRα1	GDNF-family receptor α1	Nile tilapia Dogfish Rainbow trout Common carp	[68] [60] [69] Nobrega et al. unpublished observations
NANOG	NANOG homeobox	Medaka Nile tilapia	[72] [68]
NANOS2	NANOS homolog 2	Medaka Nile tilapia Rainbow trout	[82] [68] [61]
NANOS3	NANOS homolog 3	Rainbow trout	[61]
Antibodies numbers 80 and 95	Not identified	Rainbow trout Also applied in Zebrafish Salmonids	[75] [75] [77]

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
