# Peer review of "Spermatogonial Stem Cells in Fish: Characterization, Isolation, Enrichment, and Recent Advances of In Vitro Culture Systems"

_biomolecules, 2020, doi:10.3390/biom10040644_

Round 1

Reviewer 1 Report

This manuscript entitled “Spermatogonial Stem Cells in Fish: Characterization, Isolation, Enrichment and Recent Advances of in vitro Culture Systems” describes a large amount of information regarding SSCs from basics to current status of their manipulation techniques. Overall contents were well organized and comprehensively described using the suitable and sufficient references. I think that this will be helpful for those who are interested in this field. I don’t have any query but have a minor suggestion regarding terminology. In section 3.3, you used the term “tissue culture” but I think that the culture type described in section 3.3 is “organ culture” rather than “tissue culture”. So, I recommend changing the term in later step of review process of this manuscript.

Author Response

This manuscript entitled “Spermatogonial Stem Cells in Fish: Characterization, Isolation, Enrichment and Recent Advances of in vitro Culture Systems” describes a large amount of information regarding SSCs from basics to current status of their manipulation techniques. Overall contents were well organized and comprehensively described using the suitable and sufficient references. I think that this will be helpful for those who are interested in this field. I don’t have any query but have a minor suggestion regarding terminology. In section 3.3, you used the term “tissue culture” but I think that the culture type described in section 3.3 is “organ culture” rather than “tissue culture”. So, I recommend changing the term in later step of review process of this manuscript.

Response: Thank you so much for this valuable comments. Indeed it is more accurate to use “organ culture” to describe the method in the text. We changed the term to organ culture (respectively in line 27, 46, and line 647 to line 694).

Reviewer 2 Report

The authors summarized the accumulated knowledge of SSCs and their distribution in the testis of teleosts, reviewed the primary endocrine and paracrine influence on spermatogonium self-renewal vs. differentiation in fish. To provide insight into techniques and research related to SSCs, they also reviewed available protocols and advances in enriching undifferentiated spermatogonia. They summarize in vitro germ cell culture conditions developed to maintain proliferation and survival of spermatogonia in selected fish species. Finally, they discussed tissue culture and three-dimensional models for in vitro investigation of fish spermatogenesis.

This review is timely and is helpful for the researchers in this field. Some suggestions.

  1. Figure 1A, it is appreciated to label the cell types and the cytoplasmic bridges; 1B: it would be helpful to label the two cells and other organelles;
  2. It would be easier for the readers to understand the differences in the testis histology if a cartoon is shown (line 76 to line 118);
  3. Again, it is helpful to add a cartoon in Figure 4 to explain better;
  4. Page8, a figure is needed to show the regulation.

Author Response

1. Figure 1A, it is appreciated to label the cell types and the cytoplasmic bridges; 1B: it would be helpful to label the two cells and other organelles;

Response: Thank you so much for this helpful comments. We modified the figure as the suggestion. New figure was upload with the text (line 67-77).

2. It would be easier for the readers to understand the differences in the testis histology if a cartoon is shown (line 76 to line 118);

Response: There are several review articles which we cited in this paper (including paper from co-author), showed testis structure and testicular cells with similar figures. So we introduced these papers (reference [10,15,18,19] ) from new perspective with current updates. The illustrations of testes histology would be the same, so we would like summarize these papers and guide readers of interest to read the original ones rather than show the same figure.

3. Again, it is helpful to add a cartoon in Figure 4 to explain better;

Response: There have been some papers from co-author describing spermatogonial niche and Sertoli cell with similar cartoon pictures in detail (reference [22,24,25]). In this consideration, we would like to show the real histological pictures of testis structure and spermatogonial niche in this paper, hoping to show readers a more intuitive impression.

4. Page8, a figure is needed to show the regulation.

Response: We added an illustration to explain the regulation in revised manuscript (Figure 6). It is from line 308 to line 319. It was also mentioned in text in line 272, 286, 292, 295, 300, 302 and 307.